# Next Generation Sequencing-Based Transcriptome Predicts Bevacizumab Efficacy in Combination with Temozolomide in Glioblastoma

**DOI:** 10.3390/molecules24173046

**Published:** 2019-08-22

**Authors:** Alimu Adilijiang, Masaki Hirano, Yusuke Okuno, Kosuke Aoki, Fumiharu Ohka, Sachi Maeda, Kuniaki Tanahashi, Kazuya Motomura, Hiroyuki Shimizu, Junya Yamaguchi, Toshihiko Wakabayashi, Atsushi Natsume

**Affiliations:** 1Department of Neurosurgery, Nagoya University School of Medicine, Nagoya 4668550, Japan; 2Center for Advanced Medicine and Clinical Research, Nagoya University Hospital, Nagoya 4668560, Japan

**Keywords:** glioblastoma, *IDH1* mutation, RNA-seq, NGS, GSEA, GO analysis, U87 cell line, temozolomide, bevacizumab

## Abstract

Glioblastoma (GBM), the most common and malignant brain tumor, is classified according to its *isocitrate dehydrogenase (IDH)* mutation status in the 2016 World Health Organization (WHO) brain tumor classification scheme. The standard treatment for GBM is maximal resection, radiotherapy, and Temozolomide (TMZ). Recently, Bevacizumab (Bev) has been added to basic therapy for newly diagnosed GBM, and monotherapy for recurrent GBM. However, the effect of *IDH1* mutation on the combination of Bev and TMZ is unknown. In this study, we performed transcriptomic analysis by RNA sequencing with next generation sequencing (NGS), a newly developed powerful method that enables the quantification of the expression level of genome-wide genes. Extracellular matrix and immune cell migration genes were mainly upregulated whereas cell cycle genes were downregulated in *IDH1*-mutant U87 cells but not in *IDH1*-wildtype U87 cells after adding Bev to TMZ. In vitro and in vivo studies were conducted for further investigations to verify these results, and the addition of Bev to TMZ showed a significant antitumor effect only in the *IDH1*-mutant GBM xenograft model. Further studies of gene expression profiling in *IDH1* mutation gliomas using NGS will provide more genetic information and will lead to new treatments for this refractory disease.

## 1. Introduction

Glioblastoma (GBM) is the most common and malignant brain tumor and has an extremely poor prognosis. The overall survival time (OS) of most GBM patients is less than five years after diagnosis. Even with the standard treatment of maximal resection, radiotherapy (60 Gy), and Temozolomide (TMZ, an alkylating agent), the median survival time is reported to be approximately 15 months, with less than 10% of patients surviving beyond five years [1,2,3]. Due to the development of next generation sequencing (NGS) technologies, genetic analysis for GBM has progressed rapidly in recent years. The most noteworthy development was the discovery of the *isocitrate dehydrogenase 1 (IDH1)* gene abnormality in 2008. Comprehensive genetic analysis of GBM by Parsons et al. revealed that patients with the *IDH1* mutation in GBM had a better prognosis than those without the *IDH1* mutation [4]. According to its clinical course, GBM can either be classified as primary GBM, which initially develops as GBM, and secondary GBM, which malignantly transforms from a lower grade glioma (LGG). In addition, it has been revealed that the *IDH* mutation is predominantly observed in secondary GBM and LGG [5]; thus, it is thought to be the earliest genetic event in glioma genesis [6]. Since the *IDH* mutation was recognized as an important brain tumor molecular marker, the World Health Organization revised its brain tumor classification in 2016, with GBM classified according to its *IDH* mutation status [7]. However, it remains to be elucidated which genetic alterations are involved in the malignant transformation of LGGs into secondary GBM.

RNA sequencing (RNA-seq) with NGS is a newly developed powerful method that enables the quantification of the expression level of genome-wide genes. Compared with DNA microarray, which has been widely used to evaluate the expression of a wide range of genes, RNA-seq does not require previously prepared probes, thus enabling the evaluation of unknown gene expression and the identification of unknown fusion genes or various alternative splicing. Moreover, without the risk of non-specific hybridization, it can also evaluate the expression level of transcripts of very low abundance.

In this study, we performed transcriptomic analysis by NGS and revealed that the inhibition of the vascular microenvironment played important roles in tumor growth in the *IDH1*-mutant GBM xenograft. Regarding the vascular microenvironment in GBM, there has been emerging evidence [8,9,10,11,12,13]; as GBM is known to secrete vascular endothelial growth factor (VEGF) and actively induce angiogenesis, Bevacizumab (Bev; an anti-VEGF humanized monoclonal antibody) has recently been used to treat GBM. Although Bev monotherapy for recurrent GBM has significantly increased the OS of patients [14,15,16], large-scale clinical trials (AVAglio trial; RTOG0825 trial) of newly-diagnosed GBM demonstrated that Bev in combination with standard treatment significantly prolonged progression free survival (PFS), but not OS [17,18]. However, no large-scale clinical trials or comprehensive results on the combined use of Bev and TMZ have been reported with regard to the *IDH1*-mutant GBM; therefore, the effect of *IDH1* mutations on these therapies is unknown.

## 2. Results

### 2.1. Next-Generation Sequencing Reveals Different RNA Expression of Genes Related to Immune Response and Cell Cycle in IDH1-Mutant Glioblastoma Cells but not in IDH1-Wildtype Cells after Adding Bevacizumab to Temozolomide

For two decades, DNA microarrays have been widely used to examine gene expression; however, RNA-seq, a newly developed method of NGS, has recently been replacing DNA microarrays. DNA microarrays have several disadvantages compared to RNA-seq, since they can only identify pre-defined sequences and non-specific hybridization can occur. In other words, RNA-seq can evaluate the expression levels of unknown genes and transcripts present in low abundance. Although sequencing data from NGS is enormous and more difficult to interpret, many methods of NGS data analysis have been developed. The workflow of RNA-seq data analysis used in this study is summarized in Figure 1. Gene Ontology (GO) analysis and Gene Set Enrichment Analysis (GSEA) are powerful analytical methods that assign functional terms to each gene or make various functional gene sets and calculate the statistical significance of gene sets of interest. They enable us to understand the function of genes whose expression level changed significantly after therapeutic intervention. In this study, RNA-seq revealed highly different genome-wide expression profiles in *IDH1*-mutant U87 and *IDH1*-wildtype U87 cells treated with TMZ + Bev (Appendix A). To investigate the effects of adding Bev to TMZ in those two types of cells, GO analysis and GSEA were performed. GO suggested that Bev + TMZ upregulated genes for extracellular matrix organization and immune response, but downregulated genes for cell cycle progression in *IDH1*-mutant U87 cells (Figure 2). Similarly, GSEA suggested that the combination of Bev and TMZ activated genes related to immunocyte migration (myeloid leukocyte migration, leukocyte chemotaxis, and lymphocyte migration/ chemotaxis), but suppressed genes related to the cell cycle (DNA replication, mitotic recombination, chromosome condensation, and DNA strand elongation) in *IDH1*-mutant U87 cells (Figure 3). The expression level of genes for immunocyte migration or the cell cycle was significantly different in *IDH1*-mutant U87 cells after adding Bev to TMZ (*p* = 1.68 × 10^−13^, *p* < 2.20 × 10^−22^ by Wilcoxon rank sum test, respectively; Figure 4), whereas there were no significant changes in *IDH1*-wildtype U87 cells (*p* = 0.3, *p* = 0.08 by Wilcoxon rank sum test, respectively; Figure 4). To verify these results, we performed further investigations in vitro and in vivo.

### 2.2. Bevacizumab Has no Additive Effect with Temozolomide on IDH1-Wildtype and IDH1-Mutant Glioblastoma Cells

In order to investigate the antitumor effect of Bev, TMZ, and Bev + TMZ in vitro, they were administered to *IDH1*-wildtype and *IDH1*-mutant U87 cells and the numbers of living cells were examined using Cell Counting Kit-8 (CCK-8) assays. Cells treated with 0.5% dimethyl sulfoxide (DMSO) were used as a control. As shown in Figure 5, there was no significant difference in the number of living cells in the Bev monotherapy group compared to the control group in either the *IDH1*-wildtype or *IDH1*-mutant U87 cells. When TMZ was administered, cell viability was markedly suppressed in both cells, and when treated with Bev and TMZ in combination, cell growth was remarkably suppressed in both cell lines; however, there was no significant difference between TMZ alone and Bev + TMZ.

### 2.3. Bevacizumab in Combination with Temozolomide Suppresses IDH1-Mutant Glioblastoma Xenograft

RNA-seq revealed that the addition of Bev upregulated microenvironment-related genes, including those for the extracellular matrix organization and immune response in *IDH1*-mutant GBM cells. This result agreed with the fact that there was no significant difference between *IDH1*-wiltype and *IDH1*-mutant U87 cells in vitro. Next, we investigated whether the combination of Bev and TMZ would significantly affect tumor growth in an animal model. Notably, TMZ + Bev suppressed tumor growth only in the *IDH1*-mutant U87 xenograft, whereas neither TMZ nor Bev monotherapy inhibited tumor growth in either cell type and *IDH1*-wiltype tumors did not respond to Bev + TMZ (Figure 6). These results did not contradict those of RNA-seq followed by GO and GSEA analyses.

## 3. Discussion

According to one of the largest reports of The Cancer Genome Atlas (TCGA), GBM with *IDH* mutation accounts for only a small percent of all gliomas [19]. Consequently, there has only been a few reports on specific gene expression profiling of *IDH1*-mutant GBM in comparison with that of the *IDH1*-wildtype GBM, although the *IDH1* mutation in glioma has been found to cause extensive DNA hypermethylation (glioma CpG island methylator phenotype, G-CIMP) leading to the suppression of gene expression [20]. Some studies focusing on a limited number of genes have showed that *IDH1*-mutant glioma tends to display a lower expression of programmed death ligand 1 (PD-L1), a smaller number of tumor-infiltrating lymphocytes (TILs) [21], and a lower expression of genes related to CD8^+^ cytotoxic T lymphocytes (CTL) and IFN-γ [22] than those of the *IDH1*-wildtype counterparts. Further in-depth studies on the effect of *IDH1* mutation in glioma on gene expression profiling might help to develop new therapeutic strategies for the lethal disease.

There have been some studies that have reported the effects of Bev monotherapy on cancer cells. Ramezani et al. reported that Bev resulted in apoptosis induction in parallel with the upregulation of p53, and a significant decrease in the levels of extracellular signal-regulated kinase (ERK) phosphorylation on glioblastoma cancer stem-like cells in vitro [23]. ERK phosphorylation activates genes that induce cell cycle entry and suppresses negative regulators of the cell cycle, ultimately promoting cell growth and proliferation [24,25]. Additionally, Bev induces A549 (an adenocarcinoma cell line) cell apoptosis through the mechanism of endoplasmic reticulum stress [26]. In this study, we generated a human GBM cell line transduced with the mutant *IDH1* gene (*IDH1*-mutant U87), evaluated the difference in sensitivity to currently available chemotherapy for GBM (TMZ and Bev) between the *IDH1*-mutant and *IDH1*-wildtype types, and investigated the underlying mechanism with a high-precision comprehensive analysis using RNA-seq.

IDH1 is a cytosolic enzyme that uses NADP to oxidize citrate to α-ketoglutarate (α-KG), which generates NADPH_2_ [27]. IDH2 is located in the mitochondria, but catalyzes a similar reaction. IDH3 is a part of the tricarboxylic acid cycle in the mitochondria, where NADH_2_ is oxidized to NAD^+^ in the respiratory electron transport chain to produce ATP [28,29]. While the reactions catalyzed by IDH1 and IDH2 do not yield energy, the oxidation of NADH_2_ generated by IDH3 reaction results in the production of three ATP molecules.

The *IDH1* mutation has been reported to be a novel driver mutation of gliomas [30]. It has been counterintuitive for a driver mutation as *IDH1*-mutated gliomas showed a favorable OS, in contrast to *IDH1*-mutated acute myeloid leukemia with a poor prognosis [31]. The *IDH1* mutation results in the production of considerable amounts of the oncometabolite 2-hydroxyglutarate (2-HG). The *IDH1* mutation must be present in a mutated/wildtype heterozygous allele to produce 2-HG [32]. Once citric acid is converted into isocitrate, the wildtype IDH1 enzyme converts isocitrate into α-KG via the intermediate oxalosuccinate. As this reaction constitutes oxidation, it requires NADP, which in turn is reduced to NADPH_2_. During the oxidation reaction, CO_2_ is produced. In contrast to the forward reaction, the reverse reaction is a reductive carboxylation, where CO_2_ is covalently fixed on α-KG. In turn, the mutant allele converts α-KG into 2-HG, and requires NADPH_2_ as a reduction equivalent. Thus, the generation of NADPH_2_ by cytosolic IDH1 is impaired when *IDH1* is mutated. The accumulation of 2-HG inhibits ten-eleven translocation (TET) and lysine (K)-specific demethylase (KDM) with DNA and histone demethylation activities [33,34]. The deactivation of TET and KDM leads to the accumulation of aberrant DNA and histone (such as H3K9me3 and H3K27me2) methylation [20]. Moreover, a high concentration of 2-HG is also known to inhibit α-KG-dependent dioxygenases, procollagen-proline 4-dioxygenase [35], and hypoxia-inducible factor 1α (HIF-1α) [36]. This evidence is consistent with the present study result, that is, the expression of extracellular matrix- and vascular endothelial cell-related genes was dramatically changed in *IDH1*-mutant GBM cells after adding Bev to TMZ.

In summary, by RNA-seq with NGS, we found that when Bev was added to TMZ in *IDH1*-mutant U87 cells, the expression of genes for the extracellular matrix and immune cell migration was increased, whereas that for cell cycle progression was decreased. Interestingly, there were no such variations in *IDH1*-wildtype U87 cells. In support of these results, TMZ + Bev showed an antitumor effect only in the *IDH1*-mutant GBM xenograft model in vivo. In this study, a high-precision comprehensive analysis by RNA-seq investigated the underlying mechanism of Bev efficacy to *IDH1*-mutant GBM. Further studies of gene expression profiling in *IDH1* mutation gliomas using NGS will provide more genetic information and will lead to new treatments for this refractory disease.

## 4. Materials and Methods

### 4.1. Cell Culture

The human GBM cell line U87 was transduced to express endogenous wildtype *IDH1* or the mutant *IDH1-R132H* (kindly provided from Dr. R. Pieper, University of California at San Francisco). The cell culture medium used was high glucose (4500 mg/L) Dulbecco’s Modified Eagle Medium (DMEM; Sigma-Aldrich, Co., St, Louis, MO, USA) supplemented with 10% fetal bovine serum (FBS; Sigma-Aldrich) and 1% penicillin and streptomycin (PS; Gibco^®^ by Thermo Fisher Scientific, Inc., Waltham, MA, USA), hereafter referred to as DMEM. The *IDH1*-wildtype U87 and *IDH1*-mutant U87 cell lines were cultured in DMEM in an incubator with a humidified atmosphere at 37 °C containing 5% CO_2_.

### 4.2. Therapeutic Agents In Vitro

TMZ (Sigma-Aldrich, Co., St, Louis, MO, USA) stock solution was dissolved in DMSO at 100 mM, and the final concentration of all reagents was adjusted by DMEM. There were four treatment groups for each cell line: Bev (provided from Chugai Pharmaceutical, Tokyo, Japan after the material transfer agreement) 500 ng/mL in DMSO (0.5%); TMZ 97 µg/mL in DMSO (0.5%); and Bev 500 ng/mL + TMZ 97 µg/mL in DMSO (0.5%), and DMSO (0.5%) as the control group. Hanif et al. reported that IC50 growth inhibitory effect of TMZ alone on U87 GBM cells was 0.45 mM [37], thus we used 97 µg/mL (approximately equal to 0.5 mM) of TMZ in vitro. Based on the article by Mesti et al. [38], we determined the concentration of Bev in vitro to be 500 ng/mL.

### 4.3. Therapeutic Agents In Vivo

Five-week-old nude mice were subcutaneously injected with 5 × 10^6^ cells of the *IDH1*-wildtype U87 or *IDH1*-mutant U87 cell line respectively, to prepare a subcutaneous tumor model. According to the report of Zhang et al. [39], 10 mg/kg Bev was used for our animal experiments. Previously, in the U87 xenograft model, TMZ therapeutic effects were obtained at the concentrations of 5–10 mg/kg/day [40,41]. According to these reports, we chose a TMZ dose of 7.5 mg/kg/day for 5 days. When the tumor diameter reached 5 mm, 10 mg/kg of Bev, 7.5 mg/kg of TMZ, a combination of 10 mg/kg of Bev and 7.5 mg/kg of TMZ, or DMSO (control) were administered intraperitoneally. The same amount of TMZ was intraperitoneally administered for five consecutive days. After the administration of these reagents, the tumor volume of each treatment group was measured every day.

### 4.4. Cell Viability with Cell Counting Kit-8 Assay

The *IDH1*-wildtype U87 and *IDH1*-mutant U87 cell lines were seeded at a density of 2500 cells/well in 96-well plates and cultured for 24 h with 100 µL DMEM in an incubator. The medium was then aspirated and they were treated for 120 h with their respective therapeutic agents and 100 µL DMEM in an incubator. Each treatment group used 12 wells in a 96-well plate for each cell line and 8 wells in each plate contained 100 µL DMEM without cells. At 0 and 120 h after treatment, 10 µL CCK-8 (Dojindo Molecular Technologies Inc. Kumamoto, Japan) solution was added to each well and incubated for 3 h in an incubator. Absorbance was measured using a microplate reader ARVO X4 (PerkinElmer, Inc. Waltham, MA, USA) at 450 nm. Comparative analysis was performed between the groups without the highest and lowest two measured values in each treatment group, except for the average value of the cell-free wells.

### 4.5. Therapeutic Agent Treatment for Next Generation RNA Sequencing

The *IDH1*-wildtype U87 and *IDH1*-mutant U87 cell lines were seeded at a density of 2.0 × 10^4^ cells/well in 24-well plates with 500 µL DMEM and incubated for 24 h in an incubator. They were then treated for 120 h with their respective therapeutic agents and 500 µL DMEM in an incubator, and the total RNA was isolated from each treatment group.

### 4.6. Sample Preparation for RNA Sequencing

A RNeasy Mini Kit (Qiagen Corporation, Hilden, Germany) was used to extract RNA from the *IDH1*-wildtype U87 and *IDH1*-mutant U87 cells. The quality and quantity of the RNA was evaluated using a Qubit4 Fluorometer (Invitrogen, Carlsbad, CA, USA) and a Bioanalyzer 2100 (Agilent Technologies, CA, USA). In this study, all RNA Integrity Number (RIN) scores were > 8.5. cDNA libraries for paired-end sequencing on the Hiseq 2500 platform (Illumina, San Diego, CA, USA) were prepared using a NEBNext Ultra RNA library Prep Kit for Illumina (New England Biolabs, Ipswich, MA, USA) with the NEBNext Poly(A) mRNA Magnetic Isolation Module.

### 4.7. RNA Sequence Processing

NGS reads sequenced with the Illumina Hiseq 2500 were aligned to the hg19 genome + transcriptome assembly (UCSC hg19) using TopHat v2.1.1 with the default parameters. Read counts of each gene were obtained using HTSeq [42] and differential expression was analyzed using DESeq [43]. GSEA was performed using the GSEA JAVA program (http://www.broadinstitute.org/gsea). GO analysis of genes whose expression level changed significantly (*p* < 0.05) by more than two-fold after adding Bev to TMZ in *IDH1*-mutant U87 cells was performed using http://metascape.org/.

### 4.8. Statistical Analysis

The cell viability and chemo-sensitivity data were analyzed by a Student’s *t*-test. Values of *p* < 0.05 were considered statistically significant.

## 5. Conclusions

In *IDH1*-mutant U87 cells, the expression of genes for the extracellular matrix and immune cell migration increased, whereas that of cell cycle progression decreased after the addition of Bev to TMZ. Interestingly, there were no such variations in the *IDH1*-wildtype U87 cells. In support of these results, TMZ + Bev showed an antitumor effect only in the *IDH1*-mutant U87 xenograft model in vivo. A high-precision comprehensive analysis by RNA-seq investigated the underlying mechanism of Bev efficacy to *IDH1*-mutant GBM. Further studies of gene expression profiling in *IDH1* mutation gliomas using NGS will provide more genetic information and will lead to new treatments for this refractory disease.

## Figures and Tables

**Figure 1 molecules-24-03046-f001:**
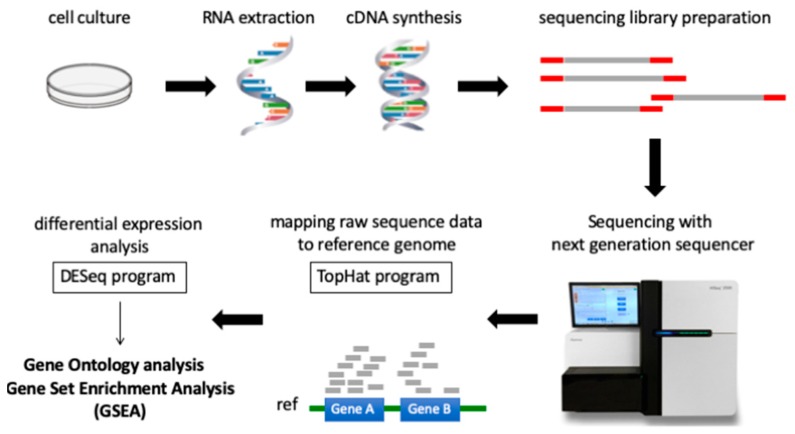
The workflow of RNA-seq and data analysis used in this study. Next Generation Sequencing reads sequenced with the Illumina Hiseq 2500 were aligned to the hg19 genome + transcriptome assembly (UCSC hg19) using TopHat v2.1.1 with the default parameters. Read counts of each gene were obtained using HTSeq and differential expression was analyzed using DESeq. Gene Ontology (GO) analysis and Gene Set Enrichment Analysis (GSEA) were performed to interpret the results.

**Figure 2 molecules-24-03046-f002:**
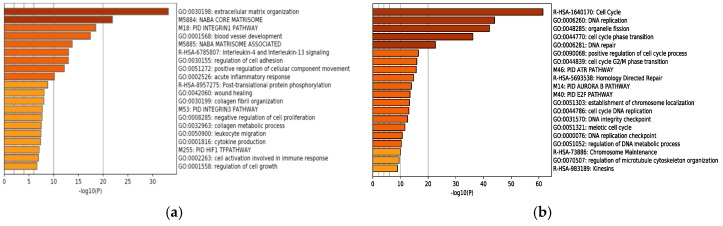
Gene Ontology analysis. (**a**) Addition of Bevacizumab (Bev) to Temozolomide (TMZ) upregulated genes for extracellular matrix organization and immune response in *IDH1*-mutant U87 cells. (**b**) Addition of Bev to TMZ downregulated genes for cell cycle progression in *IDH1*-mutant U87 cells.

**Figure 3 molecules-24-03046-f003:**
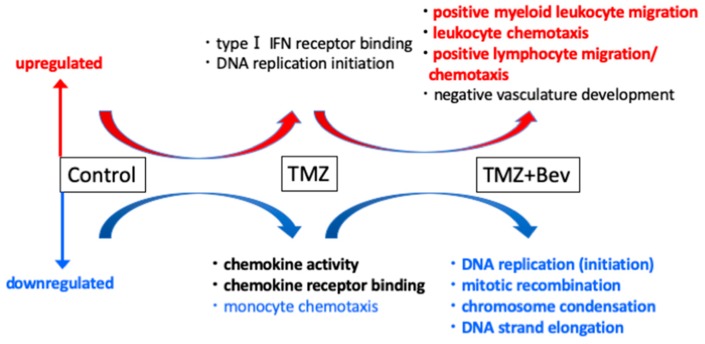
Gene Set Enrichment Analysis (GSEA) suggested that the combination of Bevacizumab and Temozolomide activated genes related to immunocyte migration (myeloid leukocyte migration, leukocyte chemotaxis, and lymphocyte migration/ chemotaxis), but suppressed genes related to the cell cycle (DNA replication, mitotic recombination, chromosome condensation, and DNA strand elongation) in *IDH1*-mutant U87 cells.

**Figure 4 molecules-24-03046-f004:**
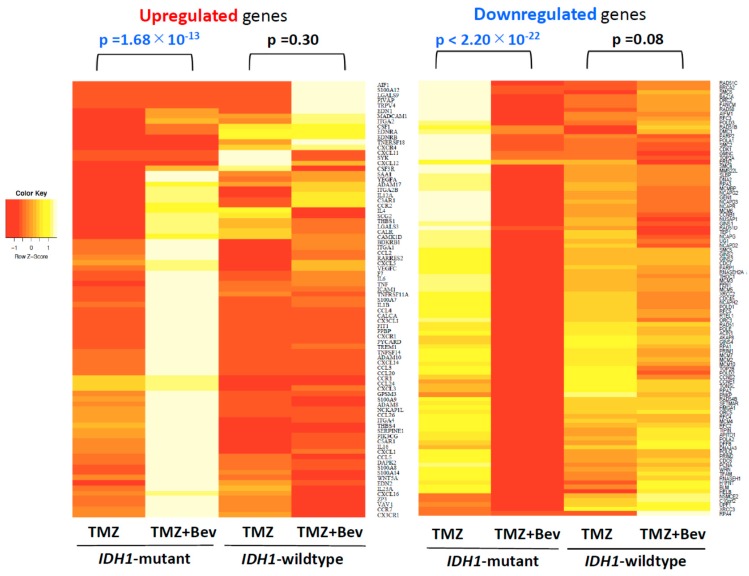
Heatmaps of upregulated genes (**left**) and downregulated genes (**right**) in *IDH1*-mutant U87 cells after adding Bevacizumab to Temozolomide. Although the expression level of genes for immunocyte migration or the cell cycle significantly changed in *IDH1*-mutant U87 cells, it showed no significant change in *IDH1*-wildtype U87 cells.

**Figure 5 molecules-24-03046-f005:**
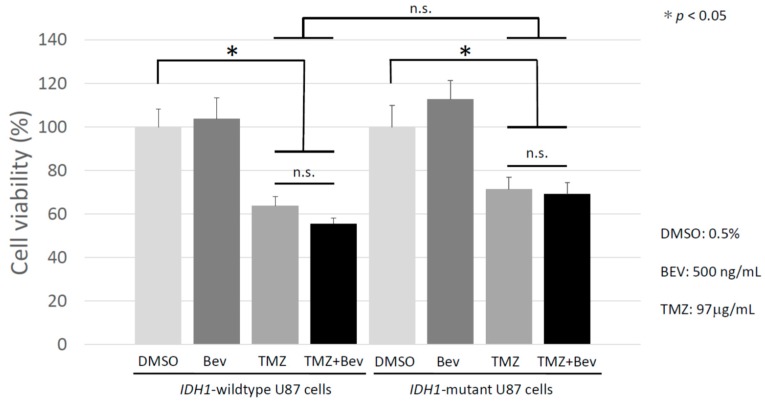
The viability of *IDH1*-wildtype U87 cells and *IDH1*-mutant U87 cells was assessed after treatment with Temozolomide (TMZ) and/or Bevacizumab (Bev). Although TMZ showed a significant antitumor effect, the addition of Bev to TMZ showed no significant add-on effect in both cell lines. **p* < 0.05; n.s., not significant.

**Figure 6 molecules-24-03046-f006:**
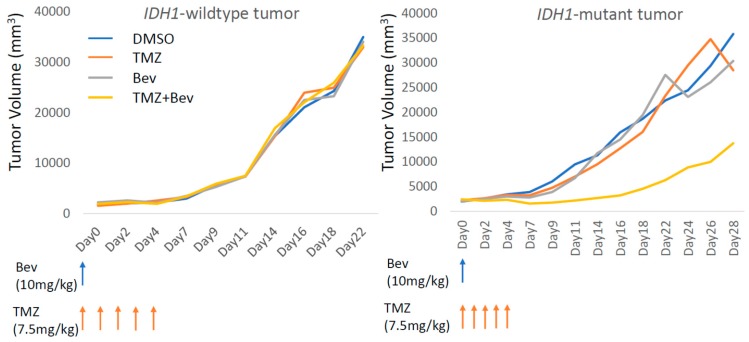
Tumor volume changes in mice xenograft models. A significant antitumor effect was observed only after Temozolomide + Bevacizumab treatment in the *IDH1*-mutant U87 xenograft model, whereas no antitumor effect was observed in the *IDH1*-wiltype U87 xenograft model.

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
