# Peer review of "Next Generation Sequencing-Based Transcriptome Predicts Bevacizumab Efficacy in Combination with Temozolomide in Glioblastoma"

_molecules, 2019, doi:10.3390/molecules24173046_

Round 1

Reviewer 1 Report

The major problem was that the authors used GO to find enriched pathways or cellular functions.  The cell type could not be defined in GO. U87 cells  are GBM brain cells. However, the enriched pathways are monocyte migration... They are immune or blood cells.  I would suggest to use IPA (or other pathway analysis program)  to search the pathways using brain or neural cells as a restricted cell type. 

Author Response

Response to reviewers

We really appreciate the reviewers’ comments and suggestions, which have been immensely beneficial to our manuscript. We have revised our paper in line with the reviewers’ suggestions as below.

To Reviewer 1

[Comment 1] The major problem was that the authors used GO to find enriched pathways or cellular functions.  The cell type could not be defined in GO. U87 cells are GBM brain cells. However, the enriched pathways are monocyte migration. They are immune or blood cells.  I would suggest to use IPA (or other pathway analysis program) to search the pathways using brain or neural cells as a restricted cell type. 

Our reply to comment 1:

We appreciate your constructive comment and suggestion. We investigated what genes were up/down-regulated in U87 GBM cells after addition of Bevacizumab (Bev). Please kindly understand that these genes were not from brain cells, but the genes that may influence the microenvironment of GBM, therefore, it is no wonder that the pathway related to immune cells came up by GO analysis.

Meanwhile, your suggestion was of interest. We reanalyzed the RNA-seq data using Metascape after excluding genes which were specific to non-brain tissues. The results were as follows (upper: upregulated, lower: downregulated genes inIDH1-mutant U87 cells after adding Bev to TMZ).

We are very pleased to confirm that the results of the downregulated genes were similar to those of the GO analysis. As for the upregulated genes, however, immune cell migration /activation was not included in the Metascape. We speculate that this was probably because the brain-focusing analysis does not include many of the immune cell migration-related genes, such as chemokine and cytokine genes. However, many literatures have reported that chemokines and cytokines play pivotal roles in inducing immune cell migration / activation in gliomas [1, 2]. Therefore, we think that GO analysis was more accurate for listing the related pathway.

Sciume, G.; Santoni, A.; Bernardini, G., Chemokines and glioma: invasion and more. J Neuroimmunol2010,224, (1-2), 8-12. Iwami, K.; Natsume, A.; Wakabayashi, T., Cytokine networks in glioma. Neurosurg Rev 2011,34, (3), 253-63; discussion 263-4.

Reviewer 2 Report

The Manuscript „Next generation sequencing-based transcriptome predicts bevacizumab efficacy in glioblastoma” is interesting. Authors should, however, consider the following issues:

Major points:

The title of the article does not include the fact, that the combination of TMZ and bevacizumab was used. The title in the current version could suggests efficacy of monotherapy with bevacizumab What is the aim of performing study on the effect of bevacizumab (anti-VEGF antibody) on cell culture, where one type of cells is used? The authors should include in the Discussion the information from other studies concerning proposed mechanism of antiproliferative effect of bevacizumab in cancer cells in culture The concentration of TMZ used on cultured cells was very high and does not correspond to the concentration used in the in vivo The authors should explain (e.g. by referring to other publications) the choice of the concentrations of the agents used in the study 

Minor points:

The authors stated, that they performed analysis of cell viability (e.g. ln 124) or proliferation (ln 215). The issues should be specified and unified within the article) Moreover the unit of concentration should be unified – why the TMZ concentration was expressed in [µM], whereas the concentration of Bevacizumab was expressed in [ng/mL] The dose of the agents in in vivo study should be expressed per body weigh The quality of Fig 4 is not sufficient. It should be improved to make it readable In the section Materials and Methods the Authors used expression: “To separate cells…” – what does it mean?

Author Response

Response to reviewers

We really appreciate the reviewers’ comments and suggestions, which have been immensely beneficial to our manuscript. We have revised our paper in line with the reviewers’ suggestions as below.

To Reviewer 2:

[Major Comment 1] The title of the article does not include the fact, that the combination of TMZ and bevacizumab was used.

Our reply to major comment 1:

This study was about the combination of TMZ and bevacizumab, not about monotherapy with bevacizumab. We apologize for the confusion. We changed the title to “Next generation sequencing-based transcriptome predicts bevacizumab efficacy in combination with temozolomidein glioblastoma”.

[Major comment 2] The concentration of TMZ used on cultured cells was very high and does not correspond to the concentration used in the in vivo The authors should explain (e.g. by referring to other publications) the choice of the concentrations of the agents used in the study 

 Our reply to major comment 2:

Thank you very much for your important comment. Hanif et al. reported that IC50 growth inhibitory effect of TMZ alone on U87 GBM cells was 0.45 mM [3]. Therefore, we used 97 µg/mL (approximately equal to 0.5 mM)of TMZ for the in vitro experiments. As for the animal experiments reported previously, in the U87 xenograft model, TMZ therapeutic effects were obtained at the concentrations of 5 -10 mg/kg/day [4, 5]. According to these reports, we chose TMZ dose as 7.5 mg/kg/day for 5 days in vivo.

Hanif, F.; Perveen, K.; Jawed, H.; Ahmed, A.; Malhi, S. M.; Jamall, S.; Simjee, S. U., N-(2-hydroxyphenyl)acetamide (NA-2) and Temozolomide synergistically induce apoptosis in human glioblastoma cell line U87. Cancer Cell Int 2014,14, (1), 133. Radoul, M.; Chaumeil, M. M.; Eriksson, P.; Wang, A. S.; Phillips, J. J.; Ronen, S. M., MR Studies of Glioblastoma Models Treated with Dual PI3K/mTOR Inhibitor and Temozolomide:Metabolic Changes Are Associated with Enhanced Survival. Mol Cancer Ther 2016,15, (5), 1113-22. Dinca, E. B.; Lu, K. V.; Sarkaria, J. N.; Pieper, R. O.; Prados, M. D.; Haas-Kogan, D. A.; Vandenberg, S. R.; Berger, M. S.; James, C. D., p53 Small-molecule inhibitor enhances temozolomide cytotoxic activity against intracranial glioblastoma xenografts. Cancer Res 2008,68, (24), 10034-9.

[Minor comments] The authors stated, that they performed analysis of cell viability (e.g. ln 124) or proliferation (ln 215). The issues should be specified and unified within the article). Moreover the unit of concentration should be unified – why the TMZ concentration was expressed in [µM], whereas the concentration of Bevacizumab was expressed in [ng/mL] . The dose of the agents in in vivo study should be expressed per body weigh. The quality of Fig 4 is not sufficient. It should be improved to make it readable.

Our reply:

Thank you for your constructive suggestions. We revised our manuscript accordingly.

[Another minor comment] In the section Materials and Methods the Authors used expression: “To separate cells…” – what does it mean? 

Our reply:

To avoid confusions, we have omitted it in the revised manuscript.

Round 2

Reviewer 1 Report

It looks like OK. Yes, brain has immune cells like macrophages. 

Author Response

Again, we really appreciate your comments and suggestions, which have been immensely beneficial to our manuscript.

Reviewer 2 Report

The Authors didn’t refer to one of the major comments from previous review:

Major points:

What is the aim of performing study on the effect of bevacizumab (anti-VEGF antibody) on cell culture, where one type of cells is used? The authors should include in the Discussion the information from other studies concerning proposed mechanism of antiproliferative effect of bevacizumab in cancer cells in culture.

Author Response

Response to reviewers

We really appreciate the reviewers’ comments and suggestions, which have been immensely beneficial to our manuscript. We have revised our paper in line with the reviewers’ suggestions as below.

To Reviewer 2:

[Major Comment] The Authors didn’t refer to one of the major comments from previous review:

Major points:

What is the aim of performing study on the effect of bevacizumab (anti-VEGF antibody) on cell culture, where one type of cells is used? The authors should include in the Discussion the information from other studies concerning proposed mechanism of antiproliferative effect of bevacizumab in cancer cells in culture.

Our reply to major comment:

We really apologize for not responding to this important major comment. The aim of investigating the effect of bevacizumab (Bev) alone on the tumor proliferation was to exclude the possibility of anti-proliferative effect of Bev on GBM cells and a GBM model in our study, because there have been reported some studies regarding the effects of Bev monotherapy on cancer cells. Ramezani et al.reported that Bev resulted in the apoptosis induction in parallel with the upregulation of p53, and a significant decrease in the levels of extracellular signal-regulated kinase (ERK) phosphorylation on glioblastoma cancer stem-like cells in vitro [1]. ERK phosphorylation activate genes that induce cell cycle entry and suppress negative regulators of the cell cycle, ultimately promoting cell growth and proliferation[2, 3].Also, Bev induced A549 (an adenocarcinoma cell line) cell apoptosis through the mechanism of endoplasmic reticulum stress [4]. Therefore, we studied the effect of bevacizumab on IDH1-mutant U87 and IDH1-wildtype U87 cells.

Ramezani, S.; Vousooghi, N.; Kapourchali, F. R.; Hadjighasem, M.; Hayat, P.; Amini, N.; Joghataei, M. T., Rolipram potentiates bevacizumab-induced cell death in human glioblastoma stem-like cells. Life Sci 2017,173, 11-19. Meloche, S.; Pouyssegur, J., The ERK1/2 mitogen-activated protein kinase pathway as a master regulator of the G1- to S-phase transition. Oncogene 2007,26, (22), 3227-39. Chambard, J. C.; Lefloch, R.; Pouyssegur, J.; Lenormand, P., ERK implication in cell cycle regulation. Biochim Biophys Acta 2007,1773, (8), 1299-310. Wang, L. L.; Hu, R. C.; Dai, A. G.; Tan, S. X., Bevacizumab induces A549 cell apoptosis through the mechanism of endoplasmic reticulum stress in vitro. Int J Clin Exp Pathol 2015,8, (5), 5291-9.